# Biologically transparent illumination is a safe, fast, and simple technique for detecting the correct position of the nasogastric tube in surgical patients under general anesthesia

**Hirofumi Hirano, Hanayo Masaki, Teppei Kamada, Yoshie Taniguchi, Eiji Masaki** *

Department of Anesthesiology, International University of Health and Welfare Hospital, Tochigi, Japan

* ejmasaki@gmail.com

**Data Availability Statement:** All relevant data are within the manuscript and its Supporting information files.

## Abstract

The aim of this study was to evaluate the effectiveness of using biologically transparent illumination to detect the correct position of the nasogastric tube in surgical patients. This prospective observational study enrolled 102 patients undergoing general surgeries. In all cases, a nasogastric tube equipped with a biologically transparent illumination catheter was inserted after general anesthesia. The identification of biologically transparent light in the epigastric area either with or without finger pressure indicated that the tube had been successfully inserted into the stomach. X-ray examination was performed to ascertain the tube position and was compared with the findings of the biologically transparent illumination technique. Biologically transparent light was detected in 72 of the 102 patients. In all of these 72 patients, the position of the nasogastric tube in the stomach was confirmed by X-ray examination. The light was not detected in the other 30 patients; X-ray examination showed that the nasogastric tube was positioned in the stomach in 21 of these 30 patients but not in the other 9. The sensitivity and specificity of the illumination were 77.4% and 100%, respectively. The results suggest that biologically transparent illumination is a useful and safe technique for detecting the correct position of the nasogastric tube in surgical patients under general anesthesia. When the BT light cannot be identified, X-ray examination is mandatory to confirm the position of the nasogastric tube.

## Introduction

Nasogastric tube (NGT) insertion is a general procedure that allows access to the stomach for diagnostic and therapeutic purposes in various clinical settings, including emergency rooms, operating rooms, intensive care units, and nursing homes. Malposition of the NGT in the respiratory tract or esophagus can lead to serious complications such as aspiration pneumonia, pneumothorax, and even death [1,2]. Several methods have been employed to aid the correct positioning of NGT. Auscultation of gurgling sounds in the epigastrium is most common method but lacks accuracy [3]. Measuring the pH of aspirates or $CO_2$ from the NGT are alternative techniques, but these techniques have lower sensitivity and specificity than X-ray examination [4,5]. Ultrasonography is another alternative but requires special training for NGT

**Funding:** The authors received no specific funding for this work.

**Competing interests:** The authors have declared that no competing interests exist.

visualization [6]. Because of the abovementioned limitations of these methods, X-ray examination is recognized as the gold standard for confirming the correct position of the NGT; however, X-ray examination also has problems, including delayed verification, radiation exposure, and the burden on medical staff [7].

Biologically transparent (BT) light is highly bio-permeable red light from a light-emitting diode (LED). BT light emitted in the abdominal digestive tract can be visualized from outside the body. Identification of BT light emitted from the tip of a special catheter (BT catheter) confirms the tip position. BT illumination has previously been utilized to determine the site for colonoscopy-assisted percutaneous sigmoidopexy [8].

The purpose of this study was to investigate whether the identification of BT light in the epigastric area from outside the body can aid in detecting the correct position of the NGT in the stomach. In other words, are the sensitivity and specificity for verifying the correct position of the NTG with BT illumination close to 100% when X-ray examination is used as the reference standard? To test this hypothesis, an NGT equipped with a BT catheter was inserted into surgical patients under general anesthesia and the BT light was checked in the epigastric area. We also performed conventional epigastric auscultation and visual inspection of gastric aspirate. X-ray examination was performed as the reference standard.

## Materials and methods

This study was approved by the ethics committee of the International University of Health and Welfare Hospital (No. 20-B-446) and conformed to the Ethical Guidelines for Clinical Studies in Japan by the Ministry of Health, Labour and Welfare as well as the principles expressed in the Declaration of Helsinki. This prospective observational study was conducted at a local university hospital with 400 beds. Verbal informed consent, which was approved by the university's institutional review board, was obtained from all patients and recorded in our database.

### BT light and BT catheter

BT light is highly bio-permeable red light emitted from an LED. The light is delivered from the light source (32037000; Neuroceuticals Inc., Tokyo, Japan; Fig 1A), which is connected to the BT catheter (Fig 1B). The BT catheter (70231000; Neuroceuticals Inc.) consists of plastic optical fibers and has a diameter of 1.5 or 2.0 mm. Because BT light is transferred to the tip of the BT catheter (Fig 1C), the NGT tip position can be identified by introducing the BT catheter into the NGT. The wavelength of the BT light is 660 nm, which passes through soft tissue but not hard tissue such as bone and cartilage. Red light similar to this wavelength is reported to pass through the human cheek but does not penetrate the skull and surrounding soft tissue [9]. BT light transferred to the tip of the BT catheter in the abdominal digestive tract can be visually confirmed from outside the body. Therefore, it seems to be possible to identify the correct tip position of the NGT in the stomach by inserting the BT catheter into the NGT and identifying the BT light emitted in the epigastric area.

### Patients and procedures

This prospective observational study enrolled patients aged 18 years or older who needed gastric decompression because of mask-bag ventilation after induction of general anesthesia for general surgeries. The decision to insert the NGT (Salem Sump™/225AABZX00046000; Covidien Japan, Tokyo, Japan) was made by the anesthesiologist. The anesthesiologist chose among three NGT sizes (14, 16, and 18 Fr) and the first 5–10 cm of the tip was lubricated. After the BT catheter was inserted, the NGT was gently advanced from the nostril to a length determined by the nose-ear-xiphoid method. The anesthesiologist did not force the insertion. If the

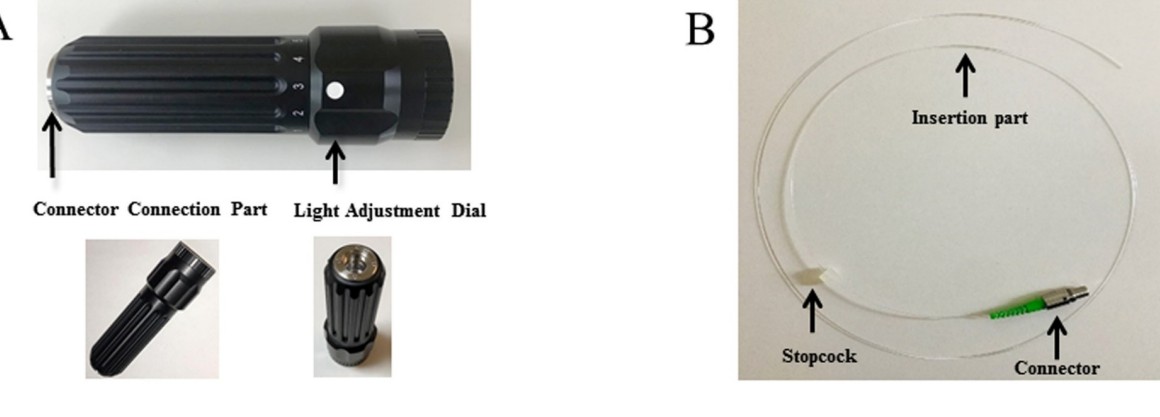

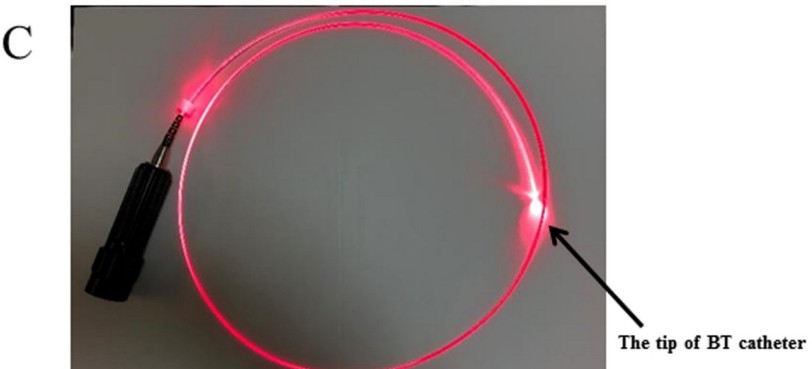

**Fig 1.** (A) The biologically transparent (BT) light source. (B) The BT catheter. (C) The BT light is transferred to the tip of BT catheter.

nasal passage was difficult, the approach was changed to the oral route. Identification of BT light was performed by the anesthesiologist either with or without finger pressure. Pressure was applied using the index finger 3–4 cm below the tenth rib on the midclavicular line in the epigastric area. After identification of the BT light, the BT catheter was removed from the NGT. Next, auscultation for gurgling sounds during injection of 10 mL of air was performed, followed by aspiration and visual inspection of gastric fluid. Finally, X-ray examination was carried out to confirm the position of the NGT. Both the anesthesiologist and the surgeon interpreted the X-ray image, and the sensitivity and specificity of BT light were calculated using the X-ray examination results as the reference standard. We also examined sensitivity and specificity in obese patients, and thickness of the abdominal wall (just below the tenth rib on the midclavicular line) was measured to distinguish the influence of obesity. Obesity was defined as BMI >27.5.

## Statistical analysis

All variables are expressed as the mean ± standard deviation. Sensitivity, specificity, positive predictive value, and negative predictive value were calculated to evaluate the diagnostic effectiveness of BT light.

We hypothesized that the NGT would always be placed in the stomach when the BT light is detected in the epigastric area from outside the body, resulting in a sensitivity and positive

predictive value of 100%. The required sample size was calculated by the Clopper-Pearson method [10] on the basis that the lower limit of the 95% confidence interval exceeds positive predictive value of 95% [11], using SAS software, version 9.4 (SAS Institute, Cary, NC). Accordingly, the procedure was performed until BT light was positively identified in 72 patients.

## Results

A total of 102 patients were enrolled. The patient characteristics are summarized in Table 1. Mean age was 63.6 ± 16.0 years and male to female ratio was 62:40. There were 19 obese patients. Mean BMI and male-to-female ratio in the obese patient group were 29.2 ± 2.5 and 14:5, respectively.

Of the 102 patients, BT light was identified in the epigastric area in 72 (Fig 2) but not in 30. X-ray examination confirmed the presence of the NGT in the stomach in 93 patients but not in the other 9. In addition, the position of the NGT in all 72 BT light-positive patients was confirmed to be in the stomach by X-ray examination. In 21 of the 30 BT light-negative patients, the presence of the NGT in the stomach was confirmed, whereas in the other 9, the NGT was found to be in the trachea (n = 1) or in the esophagus or esophago-columnar junction (n = 8) (Table 2). The sensitivity and specificity of the BT light test were 77.4% and 100%, respectively. Because the sensitivity for detecting the correct position of the NTG with BT light was not close to 100% when X-ray examination was used as the reference standard, the usefulness of BT light in identifying the correct position of the NGT was not considered equivalent to that of X-ray examination.

The abdominal thickness of true-positive and false-negative patients with BT light were 21.0 ± 8.1 (mean ± S.D.; n = 72) and 20.1 ± 5.4 (mean ± S.D.; n = 21), respectively.

The sensitivity and specificity of the auscultation test were 97.8% and 66.7%, respectively. Likewise, the sensitivity and specificity of visual inspection of the gastric aspirate were 88.1% and 55.6%, respectively.

The sensitivity and specificity of the BT light test in obese patients were 73.3% and 100%, comparable to the results in patients as a whole (Table 3).

## Discussion

The key results of this study are as follows: 1) the NGT was always positioned in the stomach when the BT light was identified in the epigastric area; 2) the BT light was not identified when the NGT was not positioned in the stomach; 3) the BT light was not always detected in the epigastric area when the NGT was positioned in the stomach. These results suggest that using BT light to identify the NGT position might be useful and safe, although further improvements in the BT illumination technique, including the quality of the equipment, are needed to reduce the false-negative rate. Currently, when BT light is not identified, X-ray examination is mandatory to confirm the position of the NGT.

**Table 1. Patient characteristics (N = 102).**

| Height (cm) | 161 ± 9 |
|---|---|
| Weight (kg) | 62 ± 14 |
| Age (years) | 64 ± 16 |
| BMI (kg/m$^2$) | 24 ± 4 |
| Percentage of obese patients | 18.6 |

Data are expressed as the mean ± standard deviation.

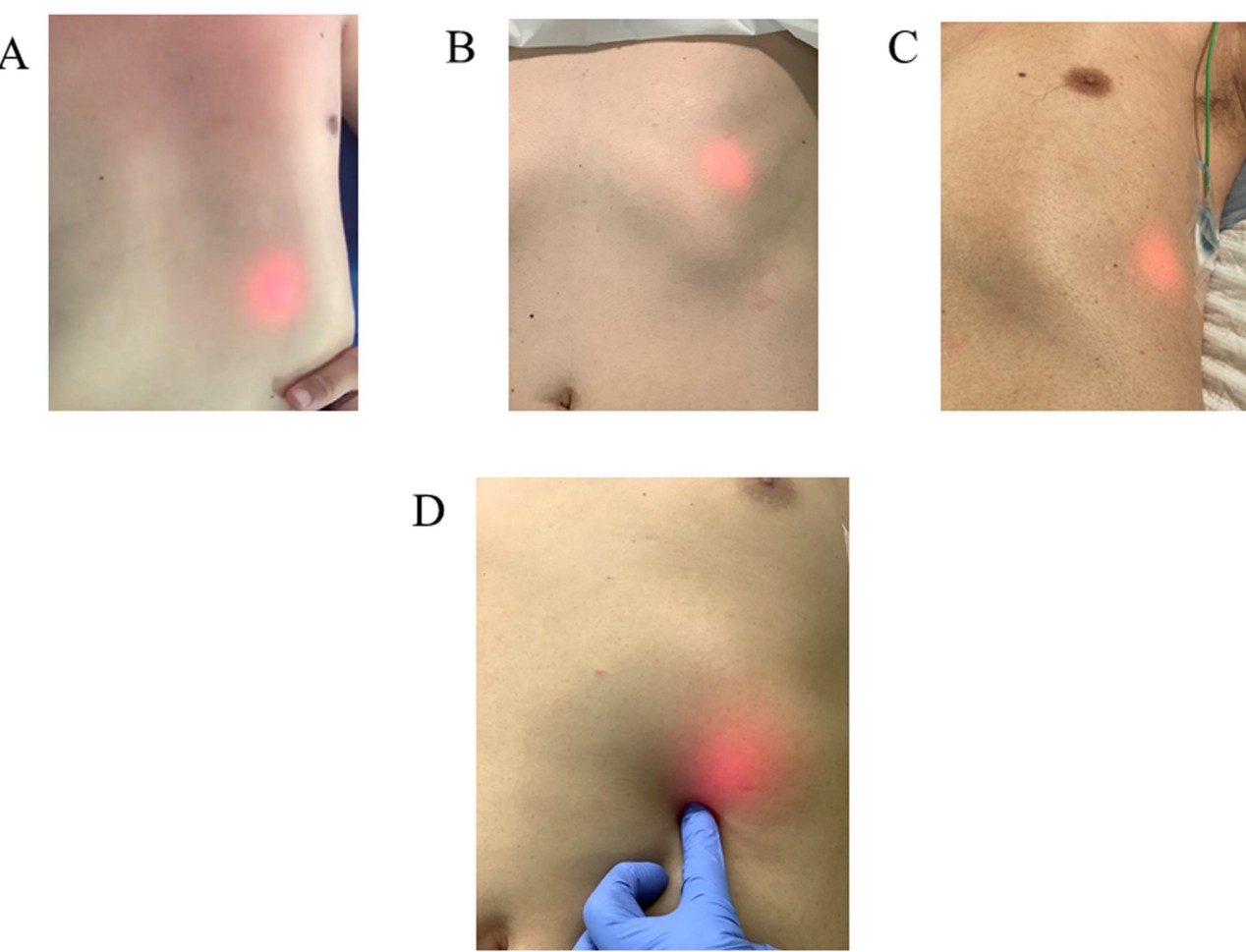

**Fig 2. Positive sign of biologically transparent (BT) light, which showed several patterns.** The BT light was detected in the epigastric area of the center site (A), the upper site (B), the lateral site (C), and using finger pressure (D).

**Table 2. Results of diagnostic tests.**

| Evaluation test | Reference test (X-ray) | | Sensitivity (%) | Specificity (%) | PPV (%) | NPV (%) |
|---|---|---|---|---|---|---|
| | Correct position (n = 93) | Incorrect position (n = 9) | | | | |
| BT light | | | | | | |
| Positive (+) | 72 | 0 | 77 | 100 | 100 | 30 |
| Negative (−) | 21 | 9 | | | | |
| Auscultation | | | | | | |
| Positive (+) | 91 | 3 | 97 | 66 | 96 | 75 |
| Negative (−) | 2 | 6 | | | | |
| Gastric aspirate | | | | | | |
| Positive (+) | 82 | 4 | 88 | 56 | 95 | 31 |
| Negative (−) | 11 | 5 | | | | |

PPV, positive predictive value; NPV, negative predictive value.

**Table 3. Results of diagnostic tests in obese patient.**

| Evaluation test | Reference test (X-ray) | | Sensitivity (%) | Specificity (%) | PPV (%) | NPV (%) |
|---|---|---|---|---|---|---|
| | Correct position (n = 15) | Incorrect position (n = 4) | | | | |
| BT light | | | | | | |
| Positive (+) | 11 | 0 | 73 | 100 | 100 | 50 |
| Negative (−) | 4 | 4 | | | | |
| Auscultation | | | | | | |
| Positive (+) | 15 | 2 | 100 | 50 | 88 | 100 |
| Negative (−) | 0 | 2 | | | | |
| Gastric aspirate | | | | | | |
| Positive (+) | 13 | 3 | 87 | 25 | 81 | 33 |
| Negative (−) | 2 | 1 | | | | |

PPV, positive predictive value; NPV, negative predictive value.

Using BT light to identify the correct position of the NGT is safe, fast, and simple. The safety of the BT illumination method was ascertained by the fact that there were no false-positive cases in the present study; that is, the specificity of this method was 100%. The NTG was always positioned in the stomach when BT light was detected in the epigastric area. Using NGT for administration of nutrition or drugs after detection of BT light in the epigastric area might avoid serious complications due to malposition of the NGT [12], although we did not specifically investigate this matter. Furthermore, no harmful events related to insertion of the BT catheter were observed in the present study. BT light was immediately identified as the NGT equipped with the BT catheter was advanced into the stomach. Little extra time (up to 60 s) is required to insert the BT catheter into the NGT, connect the BT catheter to its light source, and identify the correct position. BT light can be identified by personnel without special training or knowledge. Moreover, this method can be used in any medical setting where a BT catheter and its light source are available. This safe, fast, and simple method has the potential to solve many problems associated with NGT insertion.

This study demonstrated unsatisfactory sensitivity (77.4%) with the BT illumination technique. There are several possible reasons for this relatively low sensitivity. First, the BT light might not have been in the direction of the abdominal wall in the false-negative cases. The intensity is most powerful in the direction perpendicular to the cross-section of the BT catheter tip. Therefore, the light intensity might have been too low to be detectable from outside the body if the cross-section of the BT catheter tip was pointed away from the abdominal wall. Second, the contents of the stomach might interfere with the transmission of the BT light. Although oral intake was restricted before the procedure for all patients, a certain amount of gastric fluid was aspirated in some patients. Gastric fluid might change the transmission of BT light, thereby resulting in false-negative cases. Finally, many obese patients were included in this study. Although BT light is highly bio-permeable, the extent to which it can pass through thick tissues has not been investigated. The abdominal thickness did not differ between true-positive and false-negative patients [21.0 ± 8.1 (mean ± S.D.; n = 72) and 20.1 ± 5.4 (mean ± S.D.; n = 21), respectively]. In addition, the sensitivity and specificity of the BT light test in obese patients were comparable to the results in patients as a whole. Statistical analysis of this point cannot be performed, but it seems that abdominal thickness might not be associated with the detection of BT light in this study.

Previous studies have reported that the correct diagnostic rate (i.e., sensitivity) for the position of the NGT in the stomach is 86%-97% with ultrasonography [13–15], 78%-91% with pH

measurement of the aspirates [13,15], and 86% with $CO_2$ measurement from the NGT [4]. In contrast, when the NGT is not in the stomach, the accurate diagnostic rate (i.e., specificity) for the respective techniques was 67%-100% [13,14], 67%-86% [13,16], and 22% [4]. The present study revealed that the sensitivity and specificity when using BT illumination were 77% and 100%, respectively. These results suggest that BT illumination might be inferior to ultrasonography and pH measurement of the aspirates in terms of sensitivity. Despite the better sensitivity of these two techniques, ultrasonography and pH measurement of the aspirates have their own drawbacks, which the BT illumination method does not share. As mentioned above, special training is needed for NGT visualization with ultrasonography and it also takes more time (about 24 min compared with 1 min for BT illumination) [15]. The low specificity of pH measurement of the aspirates compared with BT illumination is a definite limitation. The present study also demonstrated that the specificity of the gastric aspirate test was quite low. Therefore, the BT illumination method might be a useful technique for identifying the correct position of the NGT. However, further improvement is needed to increase sensitivity for detecting the position of the NGT in the stomach.

There are several limitations to this study. First, the NGT was inserted under general anesthesia in the supine position. NGTs are inserted in a variety of clinical settings for various purposes. In particular, NGTs are often used in elderly nursing home residents for feeding in the sitting position without the use of sedatives. The results of the BT illumination method reported in this study might differ from cases without general anesthesia in non-supine positions. Second, we used only one type of NGT with three different diameters (14, 16, and 18 Fr). Many kinds of NGT are available in many different sizes and NGTs are made from different kinds of materials, which might affect the biological transparency of NGTs. Finally, we evaluated the usefulness of BT illumination for identifying the correct position of the NTG in only 102 patients. The number of patients was determined in order to estimate the sensitivity and positive predictive value of BT illumination compared with X-ray examination. The high specificity of BT illumination (100%) suggests that it is safe to use; however, the small number of cases in which the NGT was not positioned in the stomach (9 cases) was insufficient to confirm its complete safety. Therefore, a study involving a larger number of patients will be necessary to assess the safety of BT illumination for detecting the correct position of the NGT.

In conclusion, the NGT was always positioned in the stomach when the BT light was identified in the epigastric area, and the BT light was not identified when the NGT was not positioned in the stomach. These results suggest that using BT illumination is a safe, fast, and simple technique for identifying the position of the NGT and might be useful in surgical patients under general anesthesia. When BT light is not identified, X-ray examination is mandatory to confirm the position of the NGT.

## Supporting information

**S1 File. Study of NG tube with BT.**
(XLSX)

## Acknowledgments

We thank the operating room staff at the International University of Health and Welfare Hospital for assistance with data collection.

## Author Contributions

**Conceptualization:** Eiji Masaki.

**Data curation:** Eiji Masaki.

**Investigation:** Hirofumi Hirano, Hanayo Masaki, Teppei Kamada, Yoshie Taniguchi.

**Project administration:** Eiji Masaki.

**Writing – original draft:** Hirofumi Hirano, Eiji Masaki.

**Writing – review & editing:** Yoshie Taniguchi, Eiji Masaki.

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
