## [Decision Letter · Decision Letter 0]

1 Mar 2021

PONE-D-21-02589

Biologically transparent illumination is a safe, fast, and simple technique for detecting the correct position of the nasogastric tube in surgical patients under general anesthesia

PLOS ONE

Dear Dr. Masaki,

Thank you for submitting your manuscript to PLOS ONE. After careful consideration, we feel that it has merit but does not fully meet PLOS ONE’s publication criteria as it currently stands. Therefore, we invite you to submit a revised version of the manuscript that addresses the points raised during the review process.

Kindly address the comments made by reviewer 1 and upload all relevant data in format that will not allow to identify any of the study participants. If data can not be attached directly to the publication, kindly provide information on how and where data can be accessed by interested parties.

We look forward to receiving your revised manuscript.

Kind regards,

Benedikt Ley

Academic Editor

PLOS ONE

Journal Requirements:

3. Please clarify whether the IRB approved the use of verbal consent, and how consent was recorded.

Reviewers' comments:

Reviewer's Responses to Questions

**Comments to the Author**

1. Is the manuscript technically sound, and do the data support the conclusions?

Reviewer #1: Yes

Reviewer #2: Yes

2. Has the statistical analysis been performed appropriately and rigorously? 

Reviewer #1: Yes

Reviewer #2: Yes

3. Have the authors made all data underlying the findings in their manuscript fully available?

Reviewer #1: No

Reviewer #2: Yes

4. Is the manuscript presented in an intelligible fashion and written in standard English?

Reviewer #1: Yes

Reviewer #2: Yes

5. Review Comments to the Author

Reviewer #1: The paper would benefit from including data to support the conclusions made, including the % of obese women in the cohort; stratification of test performance by obesity; an evaluation of how BT sensitivity could be improved by combining it with auscultation and gastric pH (other point of care tools to ascertain NGT location); and a breakdown of + BT Light by anatomical site. I would recommend publication once these issues are addressed.

Minor comments include:

Page 6 – define ‘with or without finger pressure’. Where and how this is done?

Page 7 – ‘on the basis that the lower limited of the 95% confidence interval exceeds 95%’.. please clarify exceeds 95% of what?

Page 8 – Table 1. Can you provide range and % of overweight and obese patients in the cohort. One would expect performance BTL to suffer the more obese the patients are.

Page 8: ‘the power of BT light to identify the correct position of the NGT was not equivalent to that of X-ray examination’ – can you rephrase this? It’s not entirely clear what is referred to here.

Page 9: Remove ‘ the latter of which was quite low’ and expand on this in the discussion.

Page 9: Figure 2. This figure caption mentions anatomical areas where BT light was detected. Is there a breakdown of this available per anatomical site, and how this affected test performance? You are alluding to this in the first paragraph of your discussion but you are not showing the data.

Abdominal wall = epigastric area? Use consistent descriptors for your anatomical sites in the paper.

Page 11: I suggest to move the section on BTL and abdominal wall thickness to the results part of the paper, including methodology of the measurements. As above, include more details of the proportion of obese patients etc in your results. Potentially stratify results BMI.

Page 13 – discuss safety of BT light (here or in background section)

Page 12 – can a combination of BT, auscultation, and gastric aspirate (can be done in at bedside) be used to improve sensitivity?

Reviewer #2: I think it’s a clever approach.maybe more patients could have been found.

Nicely presented.Helpful for clinical practice.Simple techniques.Also nice to see the pictures at the end.

I have no more remarks about it

6. PLOS authors have the option to publish the peer review history of their article (what does this mean?). If published, this will include your full peer review and any attached files.

Reviewer #1: No

Reviewer #2: No

---

## [Author Response · Author response to Decision Letter 0]

5 Mar 2021

We responded to all comments of editor and corrected the revised manuscript according to these responses.

The responses to reviewer are in the attachment file, "Response to Reviewers"

---

## [Editor Report · Decision Letter 1]

22 Mar 2021

PONE-D-21-02589R1

Biologically transparent illumination is a safe, fast, and simple technique for detecting the correct position of the nasogastric tube in surgical patients under general anesthesia

PLOS ONE

Dear Dr. Masaki,

Thank you for submitting your manuscript to PLOS ONE. After careful consideration, we feel that it has merit but does not fully meet PLOS ONE’s publication criteria as it currently stands. Therefore, we invite you to submit a revised version of the manuscript that addresses the points raised during the review process.

We look forward to receiving your revised manuscript.

Kind regards,

Benedikt Ley

Academic Editor

PLOS ONE

Journal Requirements:

Additional Editor Comments (if provided):

On page 5 You have added the sentence: "We will open all data underlying the findings described in the present study without restriction according to the request" and this sentence is suitable for a reply to the reviewers, but should not be added to the manuscript. Kindly include all underlying data never the less.

Kindly have a native English speaker revise the introduced changes, the language in most cases will require some modification. Specifically reviewer 1 had earlier asked to revise the sentence on page 10, which you kindly revised to: "Because the sensitivity for verifying the correct position of the NTG with BT light did not close to 100% when X-ray examination is used as the reference standard, the usefulness of BT light to identify the correct position of the NGT was not equivalent to that of X-ray examination." Unfortunately this sentence is still hard to understand, kindly revise together with a native English speaker.

---

## [Author Response · Author response to Decision Letter 1]

30 Mar 2021

Reply to the Editor.

Journal Requirements:

We reviewed all the references in our manuscript. We can not find any retracted papers in our references. If you have any information about the retracted paper in our references, please let me know.

Additional Editor Comments (if provided):

On page 5 You have added the sentence: "We will open all data underlying the findings described in the present study without restriction according to the request" and this sentence is suitable for a reply to the reviewers, but should not be added to the manuscript. Kindly include all underlying data never the less.

According to the pointing out of the Editor, we delete the sentence (page 5). All underling data are included in the manuscript.

Kindly have a native English speaker revise the introduced changes, the language in most cases will require some modification. Specifically reviewer 1 had earlier asked to revise the sentence on page 10, which you kindly revised to: "Because the sensitivity for verifying the correct position of the NTG with BT light did not close to 100% when X-ray examination is used as the reference standard, the usefulness of BT light to identify the correct position of the NGT was not equivalent to that of X-ray examination." Unfortunately this sentence is still hard to understand, kindly revise together with a native English speaker.

 We are apologize for our limited fluency in English. The manuscript has been professionally edited by a native English speaker familiar with this area of research. The sentence "Because the sensitivity for verifying the correct position of the NTG with BT light did not close to 100% when X-ray examination is used as the reference standard, the usefulness of BT light to identify the correct position of the NGT was not equivalent to that of X-ray examination." was revised to “Because the sensitivity for detecting the correct position of the NTG with BT light was not close to 100% when X-ray examination was used as the reference standard, the usefulness of BT light in identifying the correct position of the NGT was not considered equivalent to that of X-ray examination. (Page 10).

 The certification of the editing is enclosed. 

Additional correction.

We are deeply sorry, but we found the simple mistakes in Table 2. The numbers of patients in Auscultation test were wrong (accordingly, sensitivity, and NPV). We corrected the numbers. Those corrections did not change any other contents in the manuscript including the conclusions at all. We are sorry again.

---

## [Editor Report · Decision Letter 2]

5 Apr 2021

Biologically transparent illumination is a safe, fast, and simple technique for detecting the correct position of the nasogastric tube in surgical patients under general anesthesia

PONE-D-21-02589R2

Dear Dr. Masaki,

We’re pleased to inform you that your manuscript has been judged scientifically suitable for publication and will be formally accepted for publication once it meets all outstanding technical requirements.

Kind regards,

Benedikt Ley

Academic Editor

PLOS ONE
---

## [Editor Report · Acceptance letter]

12 Apr 2021

PONE-D-21-02589R2 

Biologically transparent illumination is a safe, fast, and simple technique for detecting the correct position of the nasogastric tube in surgical patients under general anesthesia 

Dear Dr. Masaki:

I'm pleased to inform you that your manuscript has been deemed suitable for publication in PLOS ONE. Congratulations! Your manuscript is now with our production department. 

Kind regards, 

on behalf of

Dr. Benedikt Ley 

Academic Editor

PLOS ONE